# SCALING NON-PARAMETRIC SAMPLING WITH REPRESENTATION

## ABSTRACT

Scaling and architectural advances have produced strikingly photorealistic image generative model, yet their mechanisms remain opaque. Rather than advancing scaling, we strip away complicated engineering trick and propose a simple, non-parametric conditional generative model. Our design is grounded in three principles of natural images—(i) spatial non-stationarity, (ii) low-level regularities, and (iii) high-level semantics—and defines each pixel's distribution from its local context window. Despite its minimal architecture and no training, the model produces high-fidelity MNIST samples and visually compelling CIFAR-10 images. This combination of simplicity and strong empirical performance points toward a minimal theory of natural-image structure. The model's white-box nature also allows us to have a mechanistic understanding how the model generalize and generate diverse images. We study it by analyzing how is each pixel generated by tracing every generated pixel back to its source images. These analysis reveal a simple, compositional procedure for "part-whole generalization." These findings suggest a hypothesis for how large neural network generative model learn to generalize.

## 1 INTRODUCTION

There has been tremendous progress over the past few years in the generative model community. Early successes included parametric variants of the variational auto-encoder (VAE) [33; 45] and adversarial training with GANs [22; 43; 31]. WaveNet-style autoregressive pixel models [53] and, most recently, diffusion models [25; 32] pushed visual fidelity ever higher. Despite thousands of proposed GAN and diffusion model variants, generation quality improvements have exhibited diminishing returns, indicating limited scaling performance. On the other hand, the process of generation still remains as a black box. Many works have been dedicated to visualize [15; 57; 39], explain[5; 14; 58; 17], and simplify[1; 2] the generative models in hopes of achieving principle-first theories. A computational theory of this kind could guide the construction of simple, fully explainable generative models that still capture natural signal distributions. Related efforts toward first-principles, "white-box" representation models trace back to simpler pre-deep-learning pipelines and sparsity-based approaches [34; 52; 42; 27; 54; 56; 28]; highlight that the gap to modern deep baselines can be smaller than expected [18; 44; 7; 50]; and continue with recent principled architectures [9; 8; 37]. Complementary lines of work simplify SOTA methods [11; 13; 55; 16]; relate them to classical techniques [35; 4]; unify frameworks [4; 21; 48; 29; 26]; visualize learned representations [6; 10]; and develop theory [3; 23; 51; 4].

In this work, we take a small step toward this goal by building a minimalistic *white-box* image generative model by integrating three principles of natural images: **spatial non-stationarity, low-level regularities, and high-level semantics**. First, natural images are not statistically uniform across the frame; for example, sky and sunlight usually appear on the top of an image and objects often occupy the center while backgrounds dominate the periphery. Second, at fine spatial scales, perceptual realism depends on faithfully reproducing local cues—edges, colors, shading, and textures. Finally, global semantics and invariances—object identity, part–whole relationships, pose, viewpoint, and style—impose long-range constraints that tie distant regions together in a meaningful final product.

To realize these three aspects, we revisit Shannon's (1948) idea of generative model: *(i)* short-range context is strongly predictive, and *(ii)* sampling from empirical conditionals yields realistic data [47]. Efros and Leung [19] first brought this idea to images by treating a small patch as the "local

context" and generating pixels by copying from similar patches in real images. This method excels in stationary datasets like textures; however, it a great success in sythesizing low-level regularities such as textures.

We extend Shannon's original idea beyond just short-range context to long-range context as well. We take an autoregressive n-gram approach to generation—at each pixel we assemble a small pool of source patches that define a conditioned distribution on those three principles, then sample from it. Unlike in [19] where they define "similarity" fully based on the low-level statistics of the images, we define "similarity" based on low-level statistics, positional information, and a compact *global* representation to capture a full description of natural images. Despite its minimal architecture and no training, the model produces high-fidelity MNIST samples and visually compelling CIFAR-10 images. The simple algorithm illuminates the mechanisms of image generation process. This combination of simplicity and strong empirical performance points toward a minimal theory of natural-image structure. It also serve as a hypothesis for understanding more complex, high-performing models.

In the rest of the paper, we will introduce our method in terms of a probabilistic model. We performance ablation study to show the crucial role of each of three principles for image generation. We then evaluate our model on a standard image generation task and show the performance of the model both quantitatively and qualitatively. Finally, we present a visualization tool unique for the non-parametric model for mechanistically understand the generation procedure and generalization behaviors of the model. We discover the model is able to perform "part-whole" generalization and demonstrate the mechanism behind this generalization behavior.

## 2 METHOD: NON-PARAMETRIC GENERATIVE MODEL WITH REPRESENTATION

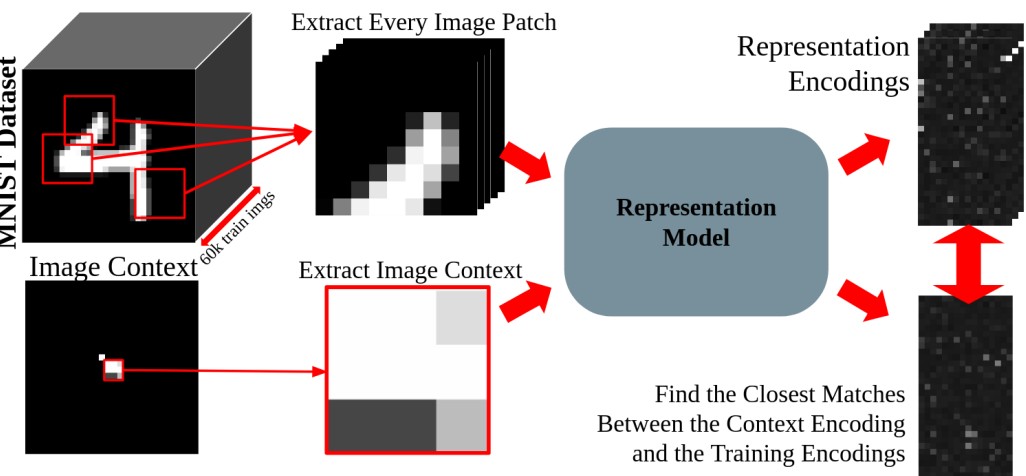

Figure 1: **Non-Parametric Sampling Conditioned on Representation.** The representation model computes some latent encoding before the last layer, and we extract that model with early exit, allowing L2-norm comparison of the resulting latent representations.

### 2.1 NON-PARAMETRIC IMAGE GENERATION

As shown in Figure 1, We model image synthesis as a conditional generative process: each pixel is sampled from an empirical conditional distribution determined by the statistics of its local context window. For a target location $p$ with context $\omega(p)$, we retrieve from the dataset the source patches most similar to $\omega(p)$ under a similarity metric $d$, and convert the center pixels of those patches into an empirical distribution over $I(p)$. Sampling from this distribution yields the next pixel; we then update the canvas and repeat until the image is complete. In what follows, we formalize the emperical distribution for a single pixel, describe the full image-generation loop, and specify the similarity metrics used to define the empirical distribution.

## 2.2 SYNTHESIZING A SINGLE PIXEL

Let $I_{\text{real}} = \{I^{(i)}\}_{i=1}^N$ be the source corpus, $I$ the image being synthesized pixel by pixel, $p \in I$ the next pixel to fill, and $\omega(p)$ the $w \times w$ context patch centered at $p$ (center unknown). We construct a candidate pool $\Omega(p; d)$ by finding patches that are similar to $\omega(p)$, based on the metric $d$. Formally,

$$\Omega(p; d) \;=\; \big\{\, \omega' \subset I_{\text{real}} : \; d\big(\omega(p), \omega'\big) \leq R \,\big\}.$$

$R$ is the threshold that decide the size of $\Omega(p; d)$. From candidate pool $\Omega(p; d)$, we form an empirical conditional distribution over the next pixel by placing equal mass on the center values of all admissible patches in the candidate pool:

$$f_p(x \mid \omega(p)) \;=\; \frac{1}{Z(p)} \sum_{\omega' \in \Omega(p;d)} \mathbf{1}\{x = c(\omega')\}, \qquad Z(p) = \big|\Omega(p; d)\big|,$$

where $c(\omega')$ denotes the center pixel of patch $\omega'$. We then sample $x \sim f_p(\cdot \mid \omega(p))$ for the pixel value of $p$.

## 2.3 SYNTHESIZING IMAGE

We grow $I$ from a small seed (e.g. a $8 \times 8$ patch randomly drawn from $I_{\text{real}}$) in concentric "shells." At each step, we choose an unfilled pixel $p$ whose neighborhood $\omega(p)$ overlaps maximally with already-synthesized pixels, then we define the empirical distribution of $p$ as $f_p(\cdot \mid \omega(p))$, we sample a pixel value to fill in the unfilled $p$. We continue this process until all pixels in $I$ are filled.

The essence of non-parametric statistics is the choice of a distance metric $d$. In the following paragraphs, we define three distance metric $d_L, d_N, d_S$ to capture three key ingredients of natural images, respectively: (i) low-level statistics, (ii) nonstationary statistics, and (iii) semantic-level statistics.

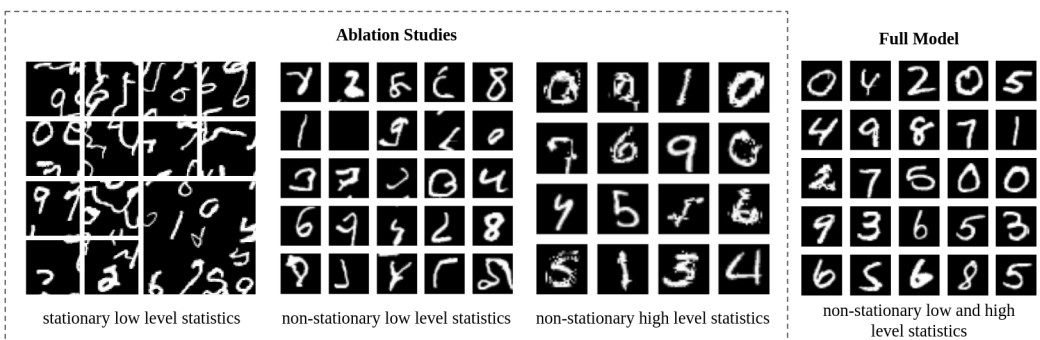

Figure 2: **Ablation Study Comparison of Statistical Components.** Each image shows the improvement from progressively adding our principle components to the generator.

## 2.4 IMAGE STATISTICS AND SIMILARITY METRICS

**Low-Level Statistics.** Low-level image statistics capture edges, shading, and textures. We use the Gaussian-weighted SSD(Sum of Squared Differences) from [19] to capture the image statistics at this level. For any candidate patch $\omega'$, define

$$d_{\text{SSD}}\big(\omega(p), \omega'\big) \;=\; \sum_x G(x) \left[\omega(p; x) - \omega'(x)\right]^2,$$

where the sum is over spatial coordinates $x$ in the $w \times w$ patch and $G(x)$ is a Gaussian weight. The candidate pool for the pixel value based on this metric is thus:

$$\Omega(p) \;=\; \Big\{\, \omega' \subset I_{\text{real}} : \; d_{\text{SSD}}\big(\omega(p), \omega'\big) \leq R_{\text{SSD}}(p) \,\Big\}.$$

Like in [19], we pick an adaptive threshold for better generation, where

$$R_{\mathrm{SSD}}(p) := (1+\epsilon) \min_{\omega'' \subset I_{\mathrm{real}}} d_{\mathrm{SSD}}\big(\omega(p), \omega''\big),$$

The SSD metric used here is identical to that of Efros and Leung[19]. Whereas Efros and Leung[19] applies this non-parametric scheme to texture synthesis with a single source image, we scale the same mechanism to image generation by using the entire dataset as the source corpus. As shown in Fig. 2, sampling with $d_{\mathrm{SSD}}$ produces patchwork artifacts—strokes and off-center fragments. The generated image resembles "texture of squiggles" rather than coherent digits. This failure is expected: $d_{\mathrm{SSD}}$ is only able to model low-level stationary statistics of natural images. However, natural image statistics are non-stationary, for example, digits are more likely to appear near the center of the frame than in the periphery. Moreover, $d_{\mathrm{SSD}}$ is purely local and cannot enforce global semantic consistency, which leads to broken strokes and misaligned fragments in the generated samples. In what follows, we refine the similarity metric $d$ to incorporate non-stationarity and semantic level statistics, which better captures image-level statistics.

**Non-Stationary and Low-Level Statistics.** We model non-stationarity natural images by limiting the candidate set to patches whose centers lie near the target pixel's location. Let $c(\omega')$ denote the center coordinate of a candidate patch $\omega'$. Define the locality distance

$$d_{\mathrm{loc}}(p, \omega') := \big\| c(\omega') - p \big\|_{\infty},$$

and fix a search radius $R_{\mathrm{loc}} > 0$. Merging this with the SSD constraint, we simply refine the same candidate set notation to require *both* conditions:

$$\Omega(p) = \Big\{ \omega' \subset I_{\mathrm{real}} : d_{\mathrm{SSD}}\big(\omega(p), \omega'\big) \le R_{\mathrm{SSD}}(p) \ \text{ and } \ d_{\mathrm{loc}}(p, \omega') \le R_{\mathrm{loc}} \Big\}.$$

Augmenting SSD with the locality constraint yields non-stationary, low-level statistics. With this refined model of spatial statistics, MNIST samples concentrate strokes near the image center, contours align, and spurious off-center fragments are markedly reduced (Fig. 2. nonstationary low-level statistics). The samples generated by modeling non-stationary statistics resemble coherent digits much more than the sample generated using only the SSD. That said, even with locality, most samples still show broken strokes or mismatched parts: non-stationary statistics alone still cannot enforce long-range semantic consistency.

**Non-Stationary, Low-Level and High-Level Statistics.** Low-level statistics alone preserve texture but not high-level structure of images. To maintain global semantic coherence, we require candidate patches to be close in a fixed, pretrained embedding (e.g., self-supervised encoders such as SimCLR; [12]). Let $\phi$ denote the SSL encoder, then

$$d_{SSL}\big(\omega(p), \omega'\big) = \big\| \phi\big(\omega(p)\big) - \phi(\omega') \big\|_2, \quad \omega' \subset I_{\mathrm{real}},$$

Merging this with the SSD constraint and locality constraint together, the final candidate set for modeling non-Stationary, low-level and high-level statistics is the following:

$$\Omega(p) = \Big\{ \omega' \subset I_{\mathrm{real}} : \begin{array}{l} d_{\mathrm{SSD}}\big(\omega(p), \omega'\big) \le R_{\mathrm{SSD}}(p) \\ \text{and } d_{\mathrm{loc}}(p, \omega') \le R_{\mathrm{loc}} \\ \text{and } d_{\mathrm{SSL}}\big(\omega(p), \omega'\big) \le R_{\mathrm{SSL}}(p) \end{array} \Big\}.$$

As with $R_{SSD}(p)$, the threshold $R_{\mathrm{SSL}}(p)$ is defined as following:

$$R_{\mathrm{SSL}}(p) := (1+\epsilon) \min_{\omega'' \subset I_{\mathrm{real}}} d_{\mathrm{SSL}}\big(\omega(p), \omega''\big),$$

Self-supervised encoders are trained to map multiple augmented views of the same image nearby in embedding space while pushing apart views of different images; as a result, $\phi$ tends to be invariant to nuisance factors (e.g., color jitter, small crops, mild deformations) and to organize features by object identity, parts, and pose. Using $d_{SSL}$ therefore injects high-level semantics that the local SSD cannot supply. Candidates must agree not only on local appearance but also on a compact, global

Table 1: **Inception Score (IS) and FID on CIFAR-10.** *Our full model (NS + L + H ) models all three image statistics: spatial non-stationarity (NS), low-level statistics (L), and high-level statistics (H). The ablated model omits low-level statistics (L).

| Model | IS ↑ | FID ↓ |
|---|---|---|
| NS + L + H * | 5.903 | 60.357 |
| NS + H | 4.081 | 32.924 |
| PixelCNN [53] | 4.60 | 65.93 |
| NCSN [49] | 8.91 | 25.32 |
| MDSM [36] | 8.31 | 31.7 |

summary of the content. As shown in Fig. 2(non-stationary high-level statistics), conditioning on the SSL encoder metric largely reduces broken strokes and misaligned fragments in the generated digits. This implies that modeling high level statistics is necessary for generating a coherent numeral. More result will be shown in next section to support this claim.

## 3 RESULTS

### 3.1 NON-PARAMETRIC GENERATION WITH REPRESENTATION

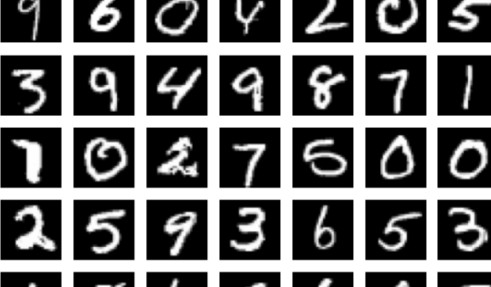
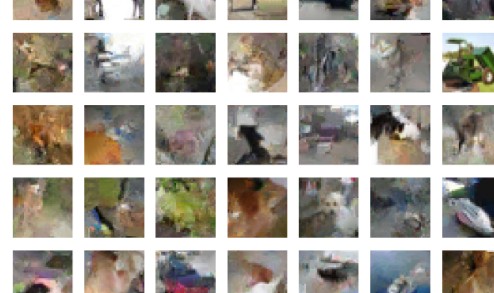

A. MNIST      B. CIFAR10

Figure 3: **Samples from Non-Parametric Image Generation.** We show the generation result for non-parametric with non-stationary, low-level and high-level statistics: MNIST (left), and CIFAR-10 (right).

**Qualitative Result.** We present samples from the proposed three-stage non-parametric generator in Figure 3. Despite occasional artifact at the finest scale, the outputs exhibit both low-level fidelity (sharp edges, coherent colors/shading and textures on CIFAR-10; continuous strokes on MNIST) and high-level structure (semantically coherent digits on MNIST; object-like layouts on CIFAR-10). Crucially, the pipeline is fully *white-box*: each pixel is drawn from an explicit empirical distribution over retrieved patches, and every choice can be traced to its source. Compared with the closest white-box baseline—Efros–Leung texture synthesis—our method produces images that are not only locally sharp but also globally consistent (see Fig. 2, Stationary Statistics is close to Efros-Leung's texture synthesis).

**Quantitative Result.** We also evaluate the generation result using FID[24] and Inception Score[46] achieves substantially lower FID on CIFAR-10 and higher Inception Score than Efros–Leung on MNIST and CIFAR-10 (Table 1).

**Class-Conditional Model.** To probe the source of our quantitative losses, we introduce a class-conditional variant that replaces latent-space conditioning with a simple restriction: retrieval is limited to patches from the target class, followed by the same locality and fine-match steps. As shown in Table 1, the class-conditional model outperforms the representation-conditioned version, indicating

Table 2: **Entropy Score.** Our full model has lower class-map entropy than the ablated representation(WO Rep), and a higher index-map entropy, which are both good. When using class conditioning(CC) instead of representation conditioning, we get zero class entropy, as there is only one class it can draw from, but a much lower index entropy due to the smaller pool of images we can select from.

| CIFAR-10 | Ours | WO Rep | CC |
|---|---|---|---|
| Class-Map | 1.075 | 1.461 | 0.0 |
| Index-Map | 2.4962 | 2.2046 | 1.550 |

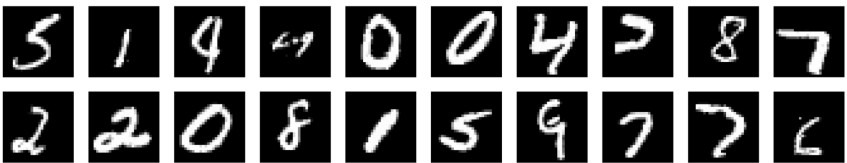

Figure 4: **MNIST Samples from Class-Conditional Non-Parametric Model.** Samples here are generated from a model with non-stationary and low-level statistics as described in Section 2.4, together with class label conditioning.

that on CIFAR-10 and MNIST much of the high-level semantics needed for generation is already captured by the class label. The gap also suggests that our current self-supervised embedding (e.g., SimCLR-style) under-represents portions of the semantic space, pointing to a straightforward avenue for improvement—stronger or task-aligned conditioning representations. Qualitatively, however (Figure 4), samples from the class-conditional and representation-conditioned models are comparable. This implies that label-free, self-supervised features are sufficient to capture the semantic structure required for visually coherent synthesis, even if they lag in measured fidelity. This label-free design keeps the model simple. Coupled with good generation result, It offers a working hypothesis for the minimal components necessary for image generation.

**Further Ablation Study.** In the method section, we provide ablation study to illustrate the importance of each component of the image statistics: non-stationary, low level an high level statistics. We show one additional ablation study by ablating low level statistic, to generate image only condition on locality constraint and SSL representation. As shown in Figure 2. non-stationary high-level statistics, removing low-level statistics cause the model to ignore high-frequency details. The ablated model generates globally correct shapes (e.g., the overall digit "3"), but the fine strokes are inconsistent or shattered, resulting images appear fragmented and locally "shattered," lacking fine-grained structural coherence. This underscores importance of the interplay between low-level and high-level statistics for modeling natural image statistics.

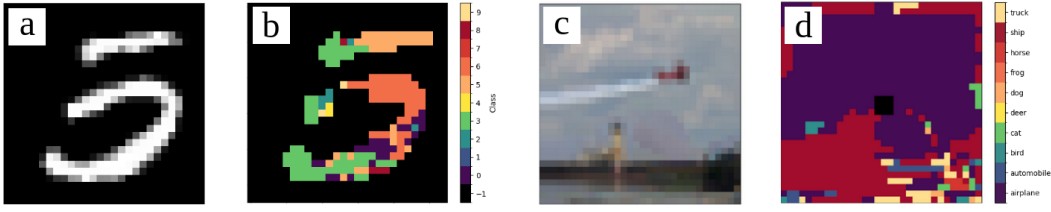

Figure 5: **Viewing Overlapping Inter-Class Features.** (a) This sample does not resemble a digit. b) The corresponding class source-map reveals conflict between classes 3, 5, and 6. c) This is another unique sample, this time with CIFAR-10. d) It's corresponding source map shows an even more interesting scenario, where two images of ship and plane are combined due to the shared sky in the background.

## 3.2 Source-Tracing: a visualization tool for interpreting generation

In our proposed non-parametric model, each pixel is produced by selecting the center of a retrieved patch from the dataset. Because every choice has a concrete source, this setting invites *mechanistic* analysis. We introduce **source-tracing**, a simple tool to expose how the model assembles images and, by analogy, to suggest what more complex neural networks might be implicitly doing.

**Source-Tracing.** At generation time, when a pixel is filled by sampling the center of some candidate patch, we log the identity of that patch's source image (and its class label), along with the source coordinates. From these logs we render two maps aligned with the generated image: (i) an *image-ID map* that colors each pixel by the source image it was copied from, and (ii) a *class map* that colors each pixel by the label of its source. These maps make the copying mechanism visible—showing which images/classes contribute where, how regions cohere semantically, and where categories overlap or blend. Figure 5 highlights two cases. In the first case (Figure 5 *a* and *b*), the sample reads as a single digit, yet the class map reveals that its left stroke is assembled from "3"-like patches while the right stroke draws from "5"/"6." This indicates that the global condition organizes *parts* (strokes, curvature) more finely than class labels and composes them into a coherent digit. In Figure 5 c, the image looks like a ship on the ocean, while the class map attributes hull and wake to "ship" sources and the upper region to sky-like patches often labeled "plane." The model thus reuses shared background structure and foreground parts from different classes to form a plausible scene. In both examples, source-tracing shows that representation-conditioning induces a part- and context-level organization richer than class identity—enabling coherent generation via recombination rather than memorization.

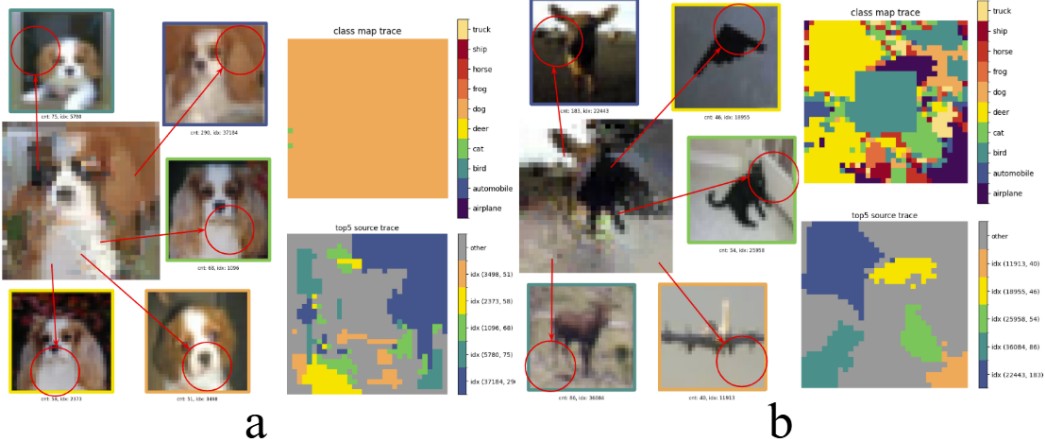

Figure 6: **Detailed Class Source-Map Trace. (a)** Center left: The beagle is the final generated image using pixels from images in the dataset. Its top five most frequently used images surrounds it. Top right: class map. Bottom right: image id Source-Map created by only these top five most frequently used images on the left, with the gray areas being from other images. **(b)** Same layout, except the representation was not invoked at generation time.

## 3.3 Mechanistic Understanding of Part-Whole Generalization

Generalization in generative models is notoriously hard to define, not to mention measuring. Unlike supervised learning, there is no single target per input, and common metrics (FID/IS) summarize distribution-level similarity to real data. The generative model could just entirely represent the training data to achieve high distribution-level similarity, without generating any "novel" sample. On the other hand, the definition of "novelty" is ambiguous. To make it more concrete, we need to specifying what kind of generalization a sample reflects—e.g., new poses, new backgrounds or novel recombinations of parts. The proposed non-parametric model, together with source-tracing, gives us a concrete lens. We observe a characteristic behavior we call part–whole generalization.

**Part–Whole Generalization.** A generated sample exhibits part–whole generalization when it builds a new whole by composing semantically coherent parts drawn from more than one training image.

Although it's hard to measure if the set of pixels used to construct the generated image is "semantically coherent." We define the following two criterion to make the definition of part–whole generalization more rigorous: (i) *class purity*—each coherent region of generated images is dominated by a single source class; (ii) *multi-image support*—within each such region, patches originate from several distinct training images (no single image contributes the majority). The second criterion ensures the sample is not a near-copy of any one training image, while the first ensures the generated image is semantically coherent. An example is shown in Figure 6, the generated image of a dog is composed of pixels from different dogs of the same breeds. The model assembles an object-level configuration it never observed as a whole, recombining familiar parts under semantic constraints rather than memorizing or producing texture-like patchworks.

**The Role of Representation in Generalization.** Figure 2 shows what happens when we remove representation from the non-parametric model and rely only on locality and modeling low-level statistics using SSD. The sampler can still stitch together locally compatible pixels (e.g., shared dark tones), but the class map becomes a patchwork of many categories: large regions are not class–pure and the result is a black blob with consistent color and shading but lacks object identity. In contrast, the beagle example in Fig. 6 (with representation conditioning) draws parts from many different dog images while remaining class–consistent across the whole object. This contrast highlights why representation is essential for part–whole generalization: it enforces criterion (i) class purity, which in turn allows criterion (ii) multi-image support to express genuine recombination rather than mere texture assembly. We quantitatively measure the both criterion with empirical entropy:

**Entropy Calculation.** As described in source-tracing section, we log, for every generated pixel, both the source image identity and its class label, yielding an *image-ID map* and a *class map* (each a 2D array of discrete labels). To quantify local diversity, we compute a sliding-window entropy over these maps: (i) form the 2D label map; (ii) slide a $7 \times 7$ window (stride 1); (iii) within each window, estimate the empirical label distribution $p(\ell)$; (iv) compute the local entropy $H = -\sum_\ell p(\ell) \log p(\ell)$; and (v) average $H$ over all windows.

Interpretation is complementary for the two maps. For class maps, low local entropy means neighboring pixels come from semantically consistent sources (good object-level coherence). For image-ID maps, high local entropy means the model assembles a region from many distinct training images (evidence of generalization at part-level rather than copying a single image). Thus, the signature of part–whole generalization is low class-entropy together with high image-ID entropy.

Empirically (Table 2), representation conditioning consistently reduces spatial class entropy relative to the model without representation conditioning, indicating more uniform, object-level structure. By contrast, a purely class-conditional variant trivially drives class entropy toward zero (all patches drawn from the target class), which is coherent but less informative. Meanwhile, the image-ID entropy remains high under representation conditioning, showing that coherent regions are composed from multiple training images rather than copied wholesale. This pattern—low class entropy, high image-ID entropy—aligns with our qualitative source-tracing and supports the claim that the model achieves true part–whole generalization.

## 4 FUTURE DIRECTIONS

We see our work as a first step towards understanding generative model by building a minimalistic white-box model. It is meant to invite the community to propose better hypothesis on how these amazing black-box generative model works. In general, we note that there's two directions for future exploration. One is to pushing the limit of generation quality with simple white-box model. The other one is to explore how to use this simple model as a hypothesis for how large black box model works.

Down this first path of simple, unsupervised and training-free white box models, our result already encouraging results, despite the bare-bones model, suggests a path to close the gap between white-box models and state-of-the-art black-box models. Potential ideas to further close the gap include: (i) replacing the current conditioning with stronger self-supervised encoders, (ii) adding light, transparent mechanisms for modeling long-range interaction (e.g., a multi-scale or attention-like retrieval that remains non-parametric), and (iii) moving from copying raw pixels to composing a small dictionary of "atomic" parts (e.g., steerable/Gabor-like elements or learned but human-readable primitives)

before rendering pixels. The goal is "simple model and good results leads to understanding": each increment should come with a clear account of what changed and why quality improved.

Another promising direction is to treat our simple non-parametric model as a working hypothesis for larger black-box generators. In spirit, this echoes recent efforts such as [30], which model reverse diffusion with an explicitly non-parametric procedure and report samples that closely resemble those from a standard diffusion model—suggesting that a complicated, multilayer network may be optimized to implement a comparatively simple algorithm. We could propose similar hypothesis for autoregressive based generative model. For example, we could encode image into token like the standard process of autoregressive model, then perform our non-parametric model in the token space, then compare the generation result with a transformer based autoregressive model. Despite the complicated multilayer and multihead attention structure, the transformer could potentially learn a simple algorithm like proposed in this paper. This perspective aligns with toy-model mechanistic interpretability work—e.g., the "Toy Model of Superposition," transformer circuits/induction heads, and grokking case studies—which aim to show that complex models can implement simple underlying algorithms [20; 40; 41; 38].

## 5  CONCLUSION

We introduced a minimal, white-box generative model by modeling three principles of natural images: spatial non-stationarity, low-level regularities, and high-level semantics—within a non-parametric, training-free framework. Despite its simplicity, the model produces diverse, realistic samples on MNIST and CIFAR-10 and exhibits generalization via part–whole composition of images rather than mere copying. Our source-tracing and entropy analyses expose the mechanism step by step, yielding a concrete understanding of how local context and global semantics interact during generation. The central takeaway is "simple model + good results = theory"; strong empirical performance from a transparent procedure points toward a minimal theory of natural-image structure. Our white-box generator offers a concrete hypothesis for the mechanisms inside black-box deep generative models. Although these large models are vastly over-parameterized, they may nevertheless discover and employ simple, compositional rules similar to those made explicit in our framework.

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

# Appendix

## A    MORE GENERATION RESULTS AND SOURCE MAP ANALYSIS

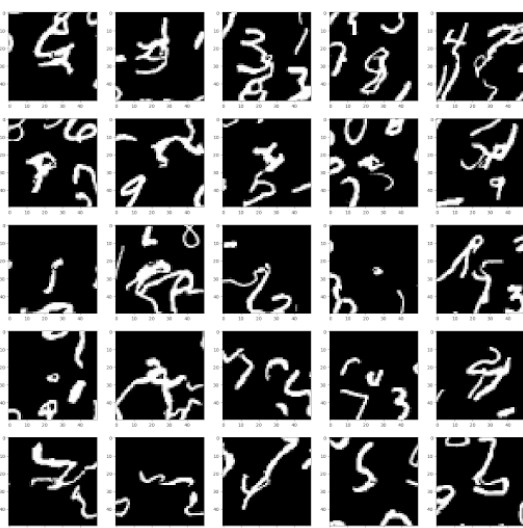

Figure 7: **Generation with Low-Level Stationary Statistics, More Examples.** Twenty-five examples of what essentially is Efros and Leung's texture synthesis applied onto MNIST. Note that the generated samples are quite large.

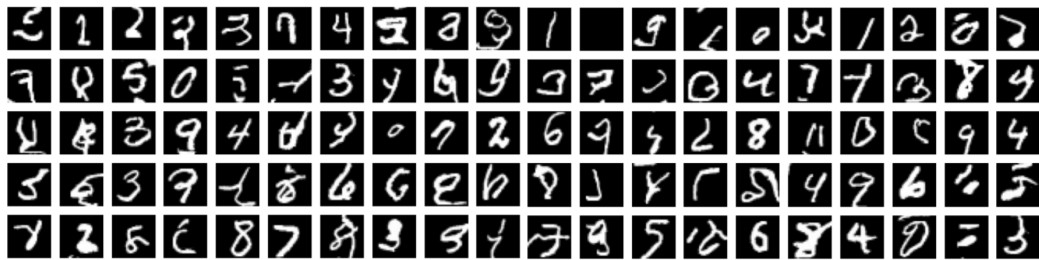

Figure 8: **Non-Stationary and low-level Statistics More Examples.** One hundred fully random examples.

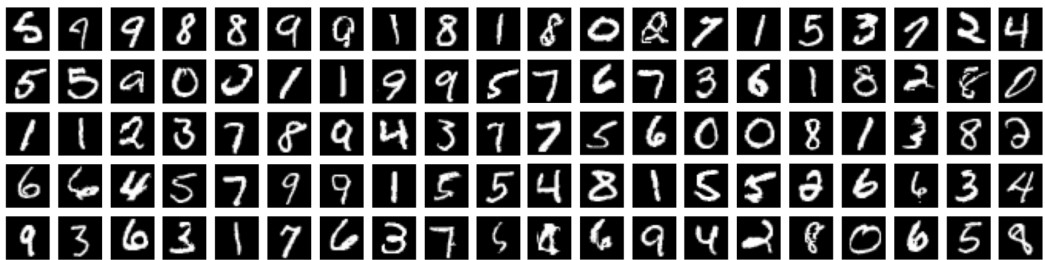

Figure 9: **Non-Stationary, low, and high-level Statistics More Examples.** For comparison to Figure 8, one hundred fully random examples.

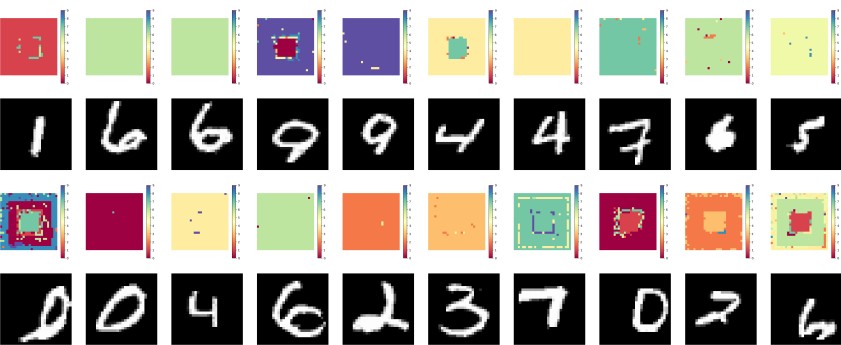

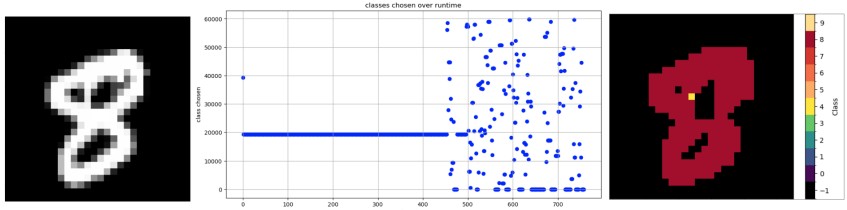

Figure 10: **MNIST Class Trace.** More examples of class tracing on MNIST generation.

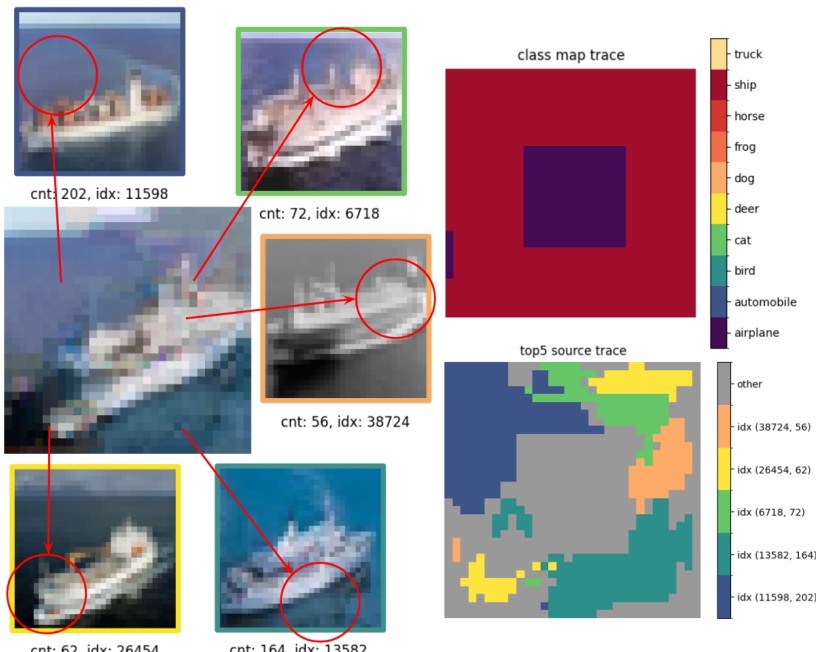

Figure 11: **Overfitting with Non-Parametric Generation.** Although ssd produced a good looking eight (left), and a unified class map (right) provides evidence of seemingly good alignment for generalization. However, The middle figure shows that nearly every single source image, except one from class 4, comes from the exact same image. The y axis the the image index (there are sixty thousand images in MNIST) and the x axis is the time stamp.

Figure 12: **Ship Source Image** Another example of class source tracing using the full model, although the center of the image started out as an airplane, but SimCLR decided it was closer to a ship. This may be a shortcoming of the representation model at small scales or incorrect window sizes, or the data may not fit the distribution of airplanes very well.

## B  LLM USAGE STATEMENT

Large Language Models were only used in the proof reading stages of this paper and played a minor role in finding and verifying references.

