# OpenReview forum: "Towards WhiteBox Generative Models: Scaling Non-Parametric Sampling with Representation"
_ICLR.cc/2026/Conference — ICLR 2026 Conference Withdrawn Submission_

### Official Review · Reviewer_zC2Z · 2025-10-18

**Soundness:** 2
**Presentation:** 2
**Contribution:** 1
**Rating:** 2
**Confidence:** 5

**Summary:**

This paper introduces an interpretable retrieval-based image generation system.
The synthetic image is being generated one pixel after another by taking the central pixel from the distribution of most similar patches contained in the given image dataset.
The proposed similarity metric takes into account both low-level and high-level patch statistics.
Notably, high-level statistics are obtained from the pretrained SimCLR neural network.

Authors show visual results om MNIST and CIFAR 10 and compute common metrics such as Inception Score and FID.

**Strengths:**

Overall, this paper introduces an original idea of non-parametric generation at small scale. The method may be useful in education purposes.

**Weaknesses:**

1. Relying on a pretrained SimCLR model to get high-level statistics seems contradicting the goal of the paper: building a  theory of natural-image structure (line 457). Since SimCLR is not interpretable, one of the components of the distance metric turns out not interpretable as well. However, no interpretable high-level statistics are proposed.
2. In Tab.1 FID of an ablated model, not including low-level statistics, it almost twice as better comparing with the full proposed model. This suggests that the diversity of generated images is higher for the ablated model. However, there is no discussion of this fact in the paper.
3. There are no results of Efros–Leung model in Tab. 1, in contrary to what is said in line 264.
4. Class-conditional model is not evaluated in Tab. 1 either, despite the claims in line 269.
5. The presented results are overly simplistic. However, I believe they can me made more interesting by incorporating hierarchical generation: from coarse scale image to high-resolution ones.

**Questions:**

I ask the authors to address the weaknesses listed above.

---

### Official Review · Reviewer_xe56 · 2025-10-27

**Soundness:** 2
**Presentation:** 2
**Contribution:** 1
**Rating:** 2
**Confidence:** 4

**Summary:**

The paper proposes a simple method to sample images that relies on simple rules about position and local statistics in both pixel space and embedding space. It extends a prior method that worked on patterns, not images, so that it can now sample images from simple datasets. The paper conducts analysis and ablations on the proposed method, enabled by its white box and non parametric nature, and claims that the method goes beyond simple memorization of the dataset.

**Strengths:**

- The method is conceptually simple and transparent, combining positional cues with pixel level and embedding space similarities.
- The non parametric design enables ablations and analyses that are difficult with black box generators.
- The work is an incremental extension that pushes pattern generation ideas toward simple images.
- The approach has potential as an auxiliary mechanism to augment generation by interpolation with a strong neural baseline, analogous to infini-gram for LLMs.

**Weaknesses:**

- The paper overclaims about natural images and neural network generalization. The evidence is limited to MNIST and CIFAR and a positional distance approach that appears tailored to centered digits or simple objects.
- The three stated principles are hand wavy and are not shown to be exhaustive or derived in a principled way and emprically only evaluated on simple datasets.
- The patch filling mechanics are unclear because with maximally overlapped patches and center placement early patches are often half unfilled, the paper does not state how the distances in pixel space and SSL space are computed for these patches. Also, near the end the last pixel cannot be centered so either centering is violated or padding is used without specification.
- Calling d_SSL a global signal is overstated because the SimCLR backbone is applied to local patches.
- Memorization remains a concern because DCR or an equivalent metric is not reported, the failure case in Figure 5 suggests a possible correlation between low DCR and visually plausible images while high DCR samples are poor, and Section 3.2, Figure 6, and Section 3.3 do not rule out reuse of common local patches in MNIST and CIFAR.
- Writing and presentation issues:
  - Line 39: current "highlight" to correct "highlighting".
  - Lines 67-68: current "We performance ablation study to show the crucial role of each of three principles ..." -> correct "We conduct an ablation study to show the crucial role of each of the three principles ...".
  - Line 71: current "for mechanistically understand" -> correct "for mechanistically understanding".
  - Lines 54-56:  The sentence "This method excels in stationary datasets like textures; however, it a great success in sythesizing low-level regularities such as textures." is not clear.
  - Line 269:  "Table 1" does not have class-conditioned. If it is referring to Table 2 then it should be clear outperform is in terms of entropy (this is discussed in another section but I am not sure what Table 1 outperforming here is referring to)

**Questions:**

- What evidence supports the claim in Lines 33-34 that generation quality improvements exhibit diminishing returns with scaling?
- How are distances computed when many pixels in a patch are unfilled, is any padding used, and how is the center handled near the end of sampling, for both pixel space and SSL distances?
- Can you report DCR or an equivalent memorization metric, add controls to rule out reuse of common local patches in MNIST and CIFAR, and test whether low DCR samples correspond to visually plausible images while high DCR samples are poor (e.g., FID of low DCR samples and FID of high DCR samples)?

---

### Official Review · Reviewer_JUXT · 2025-10-30

**Soundness:** 3
**Presentation:** 2
**Contribution:** 2
**Rating:** 4
**Confidence:** 4

**Summary:**

In this paper, the author explores non-parametric image generation by sampling each pixel from conditional distribution built from similar patches from real images. Specifically, the author encourages low-level regularities, spatial non-sationarity, and high-level semantics of the sampled pixels. The author tested their model on MNIST and CIFAR-10 generation, and show that this whitebox model provide transparency  in the generation process, allowing full tracing to the source for each generated pixel.

**Strengths:**

- The paper shows that basic image coherence can emerge from directly resampling and recombining local patches from real images, without any training. The result is limited in scope but offers a small empirical reminder of how much structure already exists in the data itself.
- The examples in Figure 2 help illustrate the effect of each similarity component (low-level, positional, semantic), although the analysis remains mostly qualitative.
- The visualization of where each generated pixel originated from offers a clear and intuitive way to understand how the model constructs images, even if the insight is primarily descriptive.
- The approach requires no optimization or learned parameters, making it easy to reproduce and inexpensive to run compared to modern generative models.

**Weaknesses:**

- The three “principles” described in L45-46  (spatial non-stationarity, low level regularities, and high-level semantics) appear to function mainly as hand-crafted inductive biases rather than theoretically grounded components.

- Experiments are restricted to MNIST and CIFAR-10, which are small and relatively low-resolution datasets. The method is only demonstrated on MNIST and CIFAR-10. While MNIST results are mostly coherent, CIFAR-10 samples appear weak in Figure 3. It is unclear whether the approach would work on more complex, high-resolution natural images.

- The method is computationally heavy due to pixel-wise nearest-neighbor search, making it impractical for larger datasets.

- Notation is confusing: The mathematics would benefit from clearer and more consistent indexed notation.
  - For example, section 2.2 $I_{real}$ as the set of real images. The same symbol $I$ is reused for the dataset, individual image.
  - Similarly, the authors use $\omega'$ and  $\omega$ for real and synthetic image patches, but $I^{(i)}$ and $I'$ for real and synthetic images, which is inconsistent index style.

- $R_{SSD}$, $R_{loc}$, $R_{SSL}$ appear to be chosen heuristically. For SSD and SSL, they include all patches within $(1+\epsilon)$ times the minimum distance, but the paper does not specify how $\epsilon$ is set or analyze sensitivity.

- The suggestion that this method could help understand modern generative models is unconvincing. The approach is purely non-parametric and relies on explicit data retrieval, while models like diffusion or transformers are parametric and synthesize from learned latent spaces. The mechanisms differ too fundamentally for this method to offer meaningful insight.

- Unclear or inconsistent writing.
  - L55–56: “This method excels in stationary datasets like textures; however, it a great success in synthesizing low-level regularities such as textures.”Grammatically incorrect and redundant; the sentence should be clarified and revised for readability.
  - L269: Mentions a “class-conditional model” in Table 1, but the table and text do not explain what this model is or what its results represent. The reference is confusing and incomplete.
  - Figure 2: The left image includes a patch that appears roughly four times larger than the others in the bottom-right corner. It is unclear whether this is intentional, an artifact of visualization, or part of the synthesis result. The caption should clarify this.

**Questions:**

- How are the thresholds chosen in practice? What is the impact of varying the thresholds?
- How large is the candidate patch pool on average, and how does this affect runtime and sample quality?
- What is the result by the “class-conditional model” in Table 1 (L269)?
- Could the authors clarify whether the “three principles” correspond to established concepts in natural image statistics, or if they are newly proposed heuristics?
- Have the authors tried the method on higher-resolution or real-world datasets to test scalability beyond MNIST and CIFAR-10?
- What is the practical utility of this approach, given its high computational cost? Since the method requires extensive nearest-neighbor searches for every pixel, how feasible is it to apply beyond small datasets or low-resolution images?
- How does the choice of patch size affect performance and image quality?

---

### Official Review · Reviewer_dfKL · 2025-10-30

**Soundness:** 2
**Presentation:** 3
**Contribution:** 2
**Rating:** 4
**Confidence:** 4

**Summary:**

The paper presents a simple and training-free probablistic model that can still create realistic iamges. It shows how small local details and overall high-level semantics work together during generation. The proposed model can generate images similar to those generated with neural networks on CIFAR10 and MNIST.

**Strengths:**

Strengths

1. The authors fomulate the 3-stage non-parametric process to uncover the mystery of generation, which is an interesting design.

2. The writeup is clear and easy to follow.

3. The ablation study shows how each components contribute to the final generation.

**Weaknesses:**

Weaknesses

1. Limited experiments. The authors only tested on CIFAR10 and MNIST, where the image is simple and low-resolution. Whether the method can scale or not still remains unclear.

2. The motivation of the paper is weak. The method built upon some discovered properties of visual clue or image. However, it's not very interesting due to two facts,

   - i). the properties are not surprised and have been incoprated in many advanced models explicitly or implicitly. Very typical case is the ResNet with CNN,

   - ii). The generation is not impressive, however introduces additional cost during gerenation, which isn't discussed in the paper.

Unclear to me how will this work guide the field towards better image generation. Like how will this affect how we build models, understand failures, etc.

**Questions:**

1. How can this work be combined with modern models?

2. What about the cases where the model collapses?

3. Any human preferences check?

---

### Note · Authors · 2025-11-14

**Comment:**

We sincerely thank the reviewers and area chair for their time and thoughtful feedback on our manuscript. After careful consideration, we have decided to withdraw the submission. Our work aims to explore a qualitatively different route to understanding generative models and image structure, prioritizing understanding over optimizing conventional benchmark-driven objectives. We believe this conceptual direction may be better suited to a different venue and therefore choose to withdraw at this time.

**Withdrawal Confirmation:**

I have read and agree with the venue's withdrawal policy on behalf of myself and my co-authors.